# Neuroprotective Effects of Ascorbic Acid, Vanillic Acid, and Ferulic Acid in Dopaminergic Neurons of Zebrafish

**DOI:** 10.3390/biomedicines12112497

**Published:** 2024-10-31

**Authors:** Fatemeh Hedayatikatouli, Michael Kalyn, Dana Elsaid, Herman Aishi Mbesha, Marc Ekker

**Affiliations:** Department of Biology, University of Ottawa, Ottawa, ON K1N 6N5, Canada

**Keywords:** neuroprotection, zebrafish, Parkinson’s disease, neurodegeneration, mitochondrial dynamics, dopamine, behavior

## Abstract

**Background/Objectives:** Parkinson’s disease (PD) is a debilitating neurodegenerative disease that targets the nigrostriatal dopaminergic (DAnergic) system residing in the human midbrain and is currently incurable. The aim of this study is to investigate the neuroprotective effects of ascorbic acid, vanillic acid, and ferulic acid in a zebrafish model of PD induced by MPTP by assessing the impact of these compounds on DAnergic neurons, focusing on gene expression, mitochondrial dynamics, and cellular stress responses. **Methods/Results:** Following exposure and qPCR and immunohistochemical analyses, ascorbic acid enhanced DAnergic function, indicated by an upregulation of the dopamine transporter (*dat*) gene and increased eGFP+ DAnergic cells, suggesting improved dopamine reuptake and neuroprotection. Ascorbic acid also positively affected mitochondrial dynamics and stress response pathways, countering MPTP-induced dysregulation. Vanillic acid only had modest, if any, neuroprotective effects on DAnergic neurons following MPTP administration. Ferulic acid exhibited the largest neuroprotective effects through the modulation of gene expression related to DAnergic neurons and mitochondrial dynamics. **Conclusions**: These findings suggest that ascorbic acid and ferulic acid can act as potential protective interventions for DAnergic neuron health, demonstrating various beneficial effects at the molecular and cellular levels. However, further investigation is needed to translate these results into clinical applications. This study enhances the understanding of neuroprotective strategies in neurodegenerative diseases, emphasizing the importance of considering interactions between physiological systems.

## 1. Introduction

Dopamine (DA) is a critical neurotransmitter in both the central and peripheral nervous systems that influences numerous behaviors ranging from movement to pleasure and memory [1]. It is well known that disruptions or perturbations to the functionality of dopaminergic (DAnergic) pathways (mesolimbic, mesocortical, tuberofundibular, and nigrostriatal) can lead to severe mental and neurodegenerative disorders [2,3]. Parkinson’s disease (PD), in particular, is a prevalent neurodegenerative disorder characterized by the degeneration of DAnergic neurons in the substantia nigra pars compacta (SNpc), leading to reduced DA levels and symptoms like bradykinesia, resting tremors, and rigidity [4]. PD arises from a combination of genetic, environmental, and mitochondrial factors [5]. Mutations in the mitochondrial genes PARKIN, PINK1, and LRRK2, as well as exposure to neurotoxins (e.g., MPTP), have been identified to contribute to its pathogenesis [6,7].

Mitochondria are essential for cellular energy production in nearly all eukaryotic cell types and regulate processes like apoptosis [8]. With midbrain DAnergic neurons being so energetically demanding, impacts to mitochondrial dynamics, such as imbalances in fission and fusion, are linked to PD and other neurodegenerative diseases that rely on midbrain DAnergic output [9]. Several proteins that play a key role in these mechanisms include Drp1 and Fis1 for fission and Mfn1/2 and Opa1 for fusion [10,11].

Over time, our understanding of PD and its etiology have advanced primarily due to leveraging various animal models in research, including the zebrafish, *Danio rerio*. Zebrafish are valuable models due to their genetic similarity to humans, transparent development, and high fecundity [12,13]. A number of tools and mutants have been developed in zebrafish to facilitate the study of DAnergic neuron biology, including the Tg(*dat:eGFP*) transgenic line, which allows for the easy visualization and study of DA neurons [14].

While no cure exists for DA neuron degeneration, a number of neuroprotective compounds have shown promise in experimental model systems. Ascorbic acid, ferulic acid, and vanillic acid have previously been documented to exhibit antioxidative and neuroprotective properties both in vitro and in vivo [15,16,17,18].

Despite widespread distribution throughout the body, ascorbic acid accumulates in high concentrations (2–10 mM) in the brain and relies on sodium vitamin C and glucose transporters to cross the blood–brain barrier [19]. Several studies have since come forward demonstrating ascorbic acid’s activity in mediating a large number of cellular processes that include reactive oxygen species detoxification [20], cholesterol metabolism [21], and hippocampal synaptic plasticity [22] and acting as a cofactor for enzymatic redox cycling [23]. Within a neurological context, recent work using zebrafish has highlighted ascorbic acid’s ability to relieve butachlor-, deltamethrin-, lead-, and neomycin-induced neurotoxicity through the observation of reduced morphological abnormalities, hatching defects, and mortality and normalized levels of expression for genetic markers indicative of oxidative stress such as *superoxide dismutase* (*sod*) and *catalase* (*cat*) [17,24,25].

Another natural phenol known to possess antioxidizing properties is ferulic acid, one of the primary active components of traditional Chinese medicines. It has been documented to influence many cellular processes that include, but are not limited to, free radical scavenging [26], lipid metabolism [27], inflammation [28], and anti-cancer signaling pathways [29]. In recent years, the scope of ferulic acid treatment has broadened to investigate its effects in neuroprotection, where ferulic acid was shown to protect rodent neurons and PC-12 cell cultures against hypoxia and ischemia-induced cytotoxicity [16,17,18,19,20,21,22,23,24,25,26,27,28,29,30].

Similarly, the use of plant-derived vanillic acid has recently come forth with therapeutic effects that attenuate the impacts of oxidative stress in a number of models and processes. Vanillic acid is produced as a byproduct from the conversion of ferulic acid into vanillin and has been shown to modulate healthy and disease states of cardiac [18], neurological [31], metabolic [32], and inflammatory [33] function.

The current study aims to investigate the neuroprotective effects of ascorbic acid, vanillic acid, and ferulic acid in an MPTP-induced model of PD using zebrafish. We hypothesize that pre-treatment with these compounds will positively impact DA neuron health. We quantified and compared differential expression of genes associated with DA biosynthesis, apoptosis, and mitochondrial dynamics and supported these gene expression data with the immunohistochemical quantification of DA neurons. The work presented in this article will be used to advance our understanding of natural phenolic compound interactions with oxidative stress and mechanisms of PD and may contribute to the identification of prospective mitigative and/or preventative measures pertaining to the pathogenesis of PD.

## 2. Materials and Methods

### 2.1. Zebrafish Care and Husbandry

All experimental procedures were approved by the University of Ottawa and adhered to the guidelines of the Canadian Council Animal Care and the Animal Care and Veterinary Service (ACVS). Adult zebrafish from the Tg(*dat:eGFP*) transgenic line [14,15,16,17,18,19,20,21,22,23,24,25,26,27,28,29,30,31,32,33,34], which express GFP specifically in DAnergic neurons, were bred to obtain larvae. Zebrafish were housed at 28.5 °C with a 14 h dark and 10 h light cycle. Embryos were maintained in E3 media with the following composition: 13 mM NaCl, 0.5 mM KCl, 0.02 mM Na_2_HPO_4_, 0.04 mM KH_2_PO_4_, 1.3 mM CaCl_2_, 1.0 mM MgSO_4_, and 4.2 mM NaHCO_3_.

### 2.2. Solution Preparation

An MPTP stock solution was prepared (MPTP; C_12_H_15_N · HCl, Product: M0896, CAS: 23007-85-4, Sigma, Oakville, ON, Canada) and diluted in distilled water to produce a concentration of 500 mM. To determine doses to be used in neuroprotection experiments, different concentration gradients for each neuroprotectant were achieved in a 6-well cell culture plate. The range of concentrations was chosen based on previous exposure studies carried out in zebrafish embryos, with ascorbic acid [17,24] and ferulic acid [35], or in *Xenopus* embryos [36]. A 10 mM stock solution of ascorbic acid was prepared using E3 embryo medium. From the stock solution, four concentration gradient doses were derived: 50 μM, 100 μM, 150 μM, and 250 μM. Likewise, a 20 mM stock solution of vanillic acid was prepared using E3 embryo medium to derive four concentration gradient doses: 250 μM, 500 μM, 750 μM, and 1000 μM. Being insoluble in water, ferulic acid was re-constituted in DMSO and further diluted to derive four concentration gradient doses: 100 μM, 150 μM, 250 μM, and 500 μM.

### 2.3. Neuroprotective Treatment and MPTP Exposure

After dose determination for neuroprotective treatment (see Section 3 below), zebrafish larvae were exposed to 50 μM ascorbic acid, 250 μM vanillic acid, and 100 μM ferulic acid solutions from 1 day post-fertilization (dpf) to 3 dpf. Subsequently, larvae were exposed to 0.25 mM MPTP from 4 dpf to 6 dpf. Exposure media were refreshed daily (Figure 1). For experiments with ferulic acid alone, larvae were exposed to 100 μM ferulic acid from 1 dpf to 3 dpf and then transferred to E3 medium until analysis at 6 dpf.

### 2.4. Swimming Activity

To monitor swimming activity following exposure, larval zebrafish from each treatment group (NTC, MPTP, ascorbic acid, vanillic acid, ferulic acid) were analyzed at 6 dpf. Twenty larvae from each group were individually placed in a 24-well plate containing E3 embryo medium. The larvae were acclimated to ambient light for 10 min before recording. Swimming activity was recorded for 10 min using the Zebralab software and the Zebrabox tracking system (https://www.viewpoint.fr/, ViewPoint Life Science, Lyon, France). The Zebrabox tracking system comprises infrared illumination, LED lights, and a mounted camera to capture swimming activity under both dark and light conditions. The behavioral parameters assessed included the total distance traveled, average velocity, and inactivity duration.

### 2.5. RNA Extraction, cDNA Synthesis, and qRT-PCR

RNA was extracted from three biological replicates, each containing 20 larvae at 6 dpf, using the TRIzol protocol (Invitrogen, Burlington, ON, Canada). RNA quality was assessed via agarose gel electrophoresis and quantified using a NanoDrop 1000 spectrophotometer (ThermoFisher Waltham, MA, USA). RNA was reverse transcribed into cDNA using the iScript™ cDNA Synthesis Kit (Bio-Rad, Mississauga, ON, Canada). qPCR was performed with SsoFast™ EvaGreen^®^ Supermix (Bio-Rad, Mississauga, ON, Canada) on a Bio-Rad CFX96 instrument. Gene expression was normalized using the comparative Cq method with reference genes *ribosomal protein L13a* (*rpl13a*) and *elongation factor 1 alpha* (*ef1a*). The primer sequences are listed in Table 1.

### 2.6. Immunohistochemistry on Whole-Mount Zebrafish Larvae

Zebrafish were euthanized with an overdose of MS-222, fixed in 4% paraformaldehyde, permeabilized with 0.1% Triton X-100, and blocked in 10% fetal bovine serum. Samples were incubated overnight with an anti-GFP antibody (1:500) and then with goat anti-mouse Alexa 488 conjugate (1:1000). After washing, samples were imaged using a Nikon, A1RsiMP Confocal microscope (Nikon, Mississauga, ON, Canada).

### 2.7. Confocal Microscopy and Image Analysis

At 6 dpf, larvae were mounted in 1% low-melting point agarose and imaged with a Nikon A1siMP confocal microscope using a 10× water immersion objective. Z-stack images were processed to create three-dimensional images. eGFP+ cell counts for clusters 8, 12, and 13 were performed using Fiji (ImageJ) software (https://imagej.net/ij/) by two independent researchers in a blinded fashion.

### 2.8. Statistical Analysis

Statistical analysis was performed using GraphPad Prism v.7. The eGFP+ cell counts were quantified from 10 to 15 zebrafish (n = 10–15), and gene expression data were collected from three pools of 20 larvae (n = 3 pools of 20 larvae). Behavioral analysis of swimming activity used sample sizes of 20 larvae, with gene expression data from two pools of 7 larvae. One-way ANOVA with Tukey’s multiple comparisons test was used for gene expression and swimming activity data. Two-way ANOVA was used for cell count number comparisons. Statistical significance was determined at *p* < 0.05 and indicated as * *p* < 0.05, ** *p* < 0.01, *** *p* < 0.001, ns = not significant (*p* > 0.05).

## 3. Results

### 3.1. Dose Determination for Neuroprotection Treatment

Given the known neurotoxicity of MPTP, we analyzed the gene expression levels, number of DAergic neurons, and behavioral phenotypes to evaluate the neuroprotective effects of ascorbic acid, vanillic acid, and ferulic acid. Neuroprotectants were administered every 24 h starting at 1 dpf, with MPTP administered to all treatment groups at 4 dpf. This timing was chosen to ensure DA neuron differentiation and blood–brain barrier (BBB) formation while preventing the inhibition of neurogenesis [37,38]. The doses chosen for neuroprotection were identified based on survival rates compared to MPTP-treated larvae. Larvae pre-treated with 50 μM ascorbic acid showed a 15% mortality rate, half that of MPTP-treated larvae (Figure 2A). Higher concentrations resulted in increased mortality. At 250 μM vanillic acid, 95% of larvae survived, which was significantly higher than MPTP-treated larvae (Figure 2B). Higher concentrations led to increased lethality, with 100% lethality at 1000 μM by 6 dpf. Treatment with 100 μM ferulic acid resulted in 100% survival, 25% higher than MPTP-treated larvae (Figure 2C). Higher concentrations led to higher lethality, with 100% lethality observed at 500 μM by 4 dpf. Thus, based on the data shown in Figure 2, the doses selected for further experiments were 50 μM for ascorbic acid, 250 μM for vanillic acid, and 100 μM for ferulic acid, as these doses showed increased survival compared to MPTP-treated larvae.

### 3.2. Protective Effect of Natural Phenolic Compounds on DA Biosynthesis Genes

We examined the impact of ascorbic acid, vanillic acid, and ferulic acid on the expression of dopamine biosynthesis genes *th1* and *dat* in zebrafish larvae exposed to MPTP (Figure 3). MPTP treatment significantly decreased *th1* and *dat* expression by 49.1% and 86.2%, respectively, compared to the control. Ascorbic acid D pre-treatment increased *th1* expression by 85% and *dat* by 8.8-fold compared to MPTP-treated larvae. Relative to the controls, *th1* showed a 5.7% decrease, and *dat* showed a 21.2% increase. Vanillic acid pre-treatment increased *th1* expression by 54% and *dat* by 3-fold compared to MPTP-treated larvae. Compared to the control, *th1* expression decreased by 21.4% and dat by 57%. Ferulic acid pre-treatment showed the most significant effects, increasing *th1* expression by 2.5-fold and *dat* by 9.9-fold compared to MPTP-treated larvae. Compared to the control, *th1* increased by 29.9% and *dat* by 36.9%. Ferulic acid alone (without MPTP) increased *th1* by 62.9% and *dat* by 71.7%. Co-treatment with MPTP reduced these increases by 52% and 48.5%, respectively.

### 3.3. Effect on Mitochondrial Fission Genes

We assessed the impact of neuroprotective agents on mitochondrial fission genes *pink1*, *parkin*, and *fis1* in zebrafish larvae exposed to MPTP (Figure 4). MPTP treatment significantly increased *pink1* (100.5%), *parkin* (305.5%), and *fis1* (49.28%) expression compared to the control. Ascorbic acid pre-treatment decreased *pink1*, *parkin*, and *fis1* expression by 35.65%, 47.5%, and 41%, respectively, compared to MPTP-treated larvae. Vanillic acid pre-treatment reduced *pink1*, *parkin*, and *fis1* expression by 13.7%, 22.8%, and 2.03%, respectively, compared to MPTP-treated larvae. Ferulic acid pre-treatment showed significant reductions in *pink1* (54.2%), *parkin* (45.4%), and *fis1* (41%) expression compared to MPTP-treated larvae. We also analyzed the effects on mitochondrial fusion genes *mfn1* and *opa1* (Figure 5). MPTP treatment decreased *mfn1* and *opa1* expression by 15% and 10%, respectively, compared to the control. Pre-treatment with ascorbic acid, vanillic acid, and ferulic acid did not significantly affect *mfn1* and *opa1* expression in contrast to mitochondrial fusion genes.

### 3.4. Effect on p53 Gene Expression

We analyzed the expression of *p53*, which is involved in DA neuron apoptosis, in zebrafish larvae exposed to MPTP (Figure 6). The MPTP treatment increased *p53* expression by 79.5% compared to the control. Ascorbic acid pre-treatment decreased *p53* expression by 34.7% compared to MPTP-treated larvae. Vanillic acid pre-treatment reduced *p53* expression by 15.7% compared to MPTP-treated larvae. Ferulic acid pre-treatment showed the most significant reduction in *p53* expression, decreasing it by 51.17% compared to MPTP-treated larvae. Overall, ascorbic acid, vanillic acid, and ferulic acid pre-treatments reduced *p53* expression, with ferulic acid showing the most pronounced effect.

### 3.5. Neuroprotective Effects of Ascorbic Acid, Vanillic Acid, and Ferulic Acid

Exposure to 0.25 mM MPTP resulted in a significant reduction in eGFP-positive neurons in the ventral diencephalon (vDC), proposed to be the teleost functional analog to the mammalian nigrostriatal pathway (Figure 7C and Figure 8A). Specifically, clusters 8, 12, and 13 showed decreases of 63%, 59.8%, and 70.7%, respectively. Overall, the vDC area exhibited a 65% reduction in eGFP-positive neurons, highlighting the neurotoxic effects of MPTP.

Pre-treatment with 50 μM ascorbic acid increased eGFP-positive neurons in clusters 8, 12, and 13 by 31%, 12.6%, and 6%, respectively, compared to the control group (Figure 7D and Figure 8B). Compared to the MPTP group, ascorbic acid pre-treatment resulted in a 3.3-fold (229.4%) increase in eGFP-positive neurons in the vDC region. Pre-treatment with 250 μM vanillic acid reduced eGFP-positive neurons in clusters 8, 12, and 13 by 34%, 41.5%, and 30.7%, respectively, compared to the control group (Figure 7E and Figure 8C). However, compared to the MPTP group, vanillic acid pre-treatment showed an 86% increase in eGFP-positive neurons in the vDC region. Pre-treatment with 100 μM ferulic acid increased eGFP-positive neurons in clusters 8, 12, and 13 by 27%, 20%, and 17%, respectively, compared to the control group (Figure 7F and Figure 8D). Compared to the MPTP group, ferulic acid pre-treatment resulted in a 3.4-fold (245.8%) increase in eGFP-positive neurons in the vDC region.

### 3.6. Ferulic Acid Alone Increases the Number of DAnergic Neurons

In the absence of MPTP, ferulic acid treatment alone (100 μM) increased eGFP-positive neurons in clusters 8, 12, and 13 by 53.7%, 33%, and 14%, respectively, compared to the control group (Figure 7G and Figure 8D). Overall, ferulic acid treatment led to a 31% increase in eGFP-positive neurons in the vDC region.

### 3.7. Effects on Locomotion

We analyzed zebrafish locomotor activity to assess the impact of neuroprotective treatments on behavior. Ferulic acid-treated larvae exhibited an average velocity of 0.125 cm/s, 1.83 times faster than MPTP-treated larvae (0.07 cm/s) (Figure 9B). Ferulic acid-treated larvae also had a shorter inactive duration, 52% lower than for MPTP-treated larvae (Figure 9C). Larvae pre-treated with ascorbic acid or with vanillic acid showed a modest trend for locomotor improvements that did not reach statistical significance. Overall, only pre-treatment with ferulic acid led to significant improvements in locomotion among the treated groups.

## 4. Discussion

This study investigated the neuroprotective effects of ascorbic acid, vanillic acid, and ferulic acid in a zebrafish model of PD induced by MPTP. Our findings demonstrate that these natural phenolic compounds offer significant protection against MPTP-induced neurotoxicity, as evidenced by improvements in gene expression related to DA biosynthesis, mitochondrial dynamics, neuronal survival, and behavioral outcomes.

### 4.1. Ascorbic Acid

Ascorbic acid has been shown to be a potent antioxidant involved in DAnergic homeostasis for decades. Under pathological conditions, the recycling capacities of ascorbic acid become significantly impaired, which acts to further amplify disease states. Several early studies implemented pre- and co-treatments of ascorbic acid using mice- and human-derived cell lines to protect against MPTP- and levodopa-induced neurotoxicity, respectively [39,40]. Ascorbic acid was not only shown to protect against neurotoxicity but also to promote the gene expression of DOPA and TH in vitro [41]. These studies highlighted the capacity of ascorbic acid to alleviate reactive oxygen species damage and to stimulate the components essential for DA function but failed to demonstrate the functional implications of neuroprotection as well as the impact on mitochondrial dynamics. A cohort study performed on elderly PD patients also documented increases in the absorption of levodopa for those with poor bioavailability when co-administered dietary ascorbic acid [42]; however, no documentation of prior supplementation to protect or prevent was suggested or implied. In our study, we showed the functional translation of ascorbic acid pre-treatment using an MPTP-induced PD model of larval zebrafish. Our results first complemented the foundational gene expression work of Seitz and colleagues through observed increases in *th1* expression relative to the MPTP-treated control, nearing levels of the NTC. The pre-treatment of ascorbic acid also protected against MPTP-induced mitochondrial fragmentation, which is mediated through *fis1* activity [43,44]. Surprisingly, the protective effects observed at the genetic level had no functional translation in locomotion. Future directions may investigate co-treatment with agents that will maintain molecular stability to prevent oxidization or to promote ascorbic acid recycling through increasing the reduction rate of dehydroascorbic acid back into ascorbic acid [45].

### 4.2. Ferulic Acid

The therapeutic effects of ferulic acid on neurobiology were first thoroughly investigated in the context of Alzheimer’s disease [46,47,48]. However, over the past decade, more attention has shifted to exploring mechanisms of neuroprotection in neurotoxin-induced models of PD but has failed to do so using zebrafish. Here, we demonstrate the first protective effects of ferulic acid pre-treatment on DAnergic and mitochondrial homeostatic gene expression, as well as motor performance following MPTP exposure using zebrafish larvae. In 6-OHDA lesioned mice, ferulic acid preserved mitochondrial dynamics through suppressing fission-related *Drp* expression while maintaining fusion-related *Mfn2* expression. The free radical scavenging and antioxidizing effects also resulted in comparable *Th* mRNA levels to the sham control [49]. Despite no recorded effects on behavior following 6-OHDA lesioning, we found ferulic acid pre-treatment to be similarly efficacious in mitigating the MPTP-induced downregulation of *th1* and *fis1* gene expression in zebrafish. Regarding motor effects, our results were corroborated by murine studies which found that ferulic acid improved the locomotor consequence of MPTP by enhancing performance during rotarod and swimming assays. Interestingly, these protective effects were abolished in *nuclear factor E2-related factor 2* (*Nrf2*)-null mice, suggesting that the MAPK activation of the Nrf2 pathway involved in antioxidation is responsible for ferulic acid counteracting MPTP-induced oxidative stress [50]. This pathway was previously explored in a complementary neuroprotective study using the bioflavonoid pinustrobin in zebrafish larvae where Li and colleagues reported restored TH+ immunoreactivity and swimming activity following prolonged Nrf2 and PI3K/ATK activation relative to MPTP-treated controls. Given the shared activation of Nrf2 following ferulic acid and pinustrobin pre-treatments in an MPTP-induced PD model using zebrafish, we can postulate that Nrf2 signaling is the driving force behind the observed neuroprotective effects on locomotion, in addition to gene expression; however, further studies are needed to confirm this hypothesis.

### 4.3. Vanillic Acid

Mitochondrial dysfunction plays an important role in the pathogenesis of various neurodegenerative disorders including PD. Recent studies have begun to explore different phytochemicals that can reduce oxidative stress damage, and vanillic acid has become increasingly popular due to its free scavenging properties. In SH-SY5Y cells, vanillic acid augmented mitochondrial biogenesis while also increasing the expression of genes associated with the synthesis and differentiation of DAnergic neurons (TH, *nuclear receptor subfamily 4 group A member 2* (*NR4A2* or *NURR1*), and *solute carrier family 6 member 3* (*SLC6A3*)) [51]. Conversely, vanillic acid was shown to negatively impact the transcript levels of genes associated with early-onset PD such as *LRRK2* [52]. These effects contrast those observed in our MPTP-induced model of neurodegeneration, where we found that vanillic acid failed to improve mitochondrial fission states as shown by *fis1* expression, while also exhibiting decreased DA production through reduced *th1* mRNA levels. Vanillic acid neuroprotection has previously been unexplored in the context of MPTP; however, a similar outcome was observed in a rotenone-induced model of neurotoxicity using rats. Sharma and colleagues showed that co-treatment of vanillic acid with rotenone showed improvements in locomotion and behavior at low doses without altering DA levels in the brain relative to the rotenone-treated group [52]. Higher doses did, however, improve the therapeutic effect of vanillic acid. Given the similar mechanism of oxidative stress between rotenone and MPTP targeting complex I of the mitochondrial electron transport chain, it may be promising to explore increased concentrations administrated through intraabdominal, intracerebral, or cerebroventricular injections in an adult zebrafish model to enhance the neuroprotection of vanillic acid on DA neurotransmission and mitochondrial homeostasis.

### 4.4. Correlative and Comparative Effects of the Three Neuroprotectants

For each of the three potential neuroprotectants that were tested, there was a clear correlation between the effects that each compound had on DAnergic neuron numbers (Figure 8), expression of genes involved in DAnergic function (Figure 3), expression of mitochondrial genes (Figure 4), and p53 expression (Figure 6). Thus, pre-treatment with ferulic acid consistently reverted the effects of MPTP treatment on these parameters. Pre-treatment with ascorbic acid consistently reverted, at least in part, the effects of MPTP, while pre-treatment with vanillic acid either had no impact on the MPTP-generated effects or had minor effects that did not reach statistical significance. Ferulic acid pre-treatment clearly had the strongest impact. The lack of a significant effect of vanillic acid contrasts with the actions of this chemical in cell culture or rat models, as mentioned in the preceding section. Unfortunately, increasing the concentration of vanillic acid in our experiments led to lethality (Figure 2). The mechanistic differences between the three compounds in the context of MPTP-induced neurotoxicity are unclear at this time, although some clues are suggested in the previous sections.

## 5. Conclusions

Given the complexity of PD pathogenesis, it is imperative to continue efforts to elucidate natural compounds that can be leveraged to alleviate various cellular symptoms, such as oxidative stress. Our study demonstrates that ascorbic acid, vanillic acid, and ferulic acid exhibit significant neuroprotective effects in an MPTP-induced zebrafish model of PD. Ferulic acid, in particular, showed the most robust effects across multiple parameters, including gene expression, neuronal survival, and behavior. Future research should focus on identifying the precise mechanisms that underlie these protective effects to determine the translational feasibility of recapitulating these effects in living mammalian models and to better understand the interactions between these compounds and existing PD treatments (L-DOPA) that can potentially pave the way for combinatorial therapeutic approaches to treating PD. The findings reported in this study underscore the importance of exploring natural compounds in the context of degenerative diseases and further demonstrates the potential of ascorbic acid, vanillic acid, and ferulic acid to alleviate physical and cellular perturbances of PD.

## Figures and Tables

**Figure 1 biomedicines-12-02497-f001:**
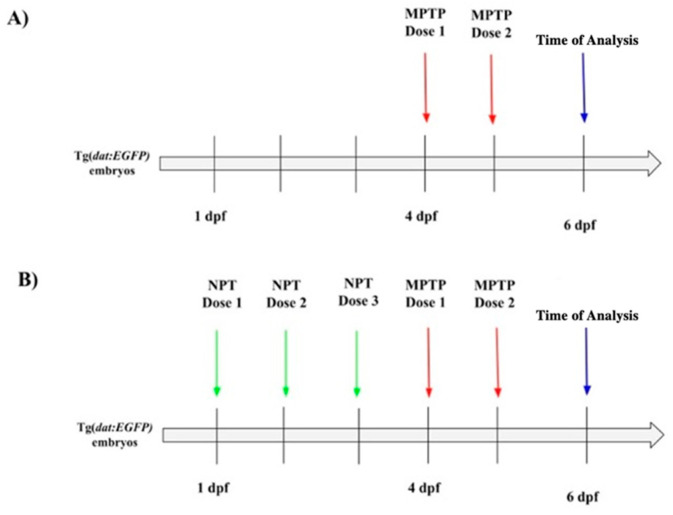
Timeline of the neuroprotective treatment and MPTP neurotoxin exposure in zebrafish larvae. (**A**) MPTP treatment: From the time of hatching until 4 days post-fertilization (dpf), the larvae were maintained in E3 Embryo medium. Subsequently, two doses of MPTP were administered: one dose at 4 dpf and another dose at 5 dpf. (**B**) Neuroprotective treatments (NPTs): In this group, neuroprotective doses (NPTs) were administered daily at 1 dpf, 2 dpf, and 3 dpf. At 4 dpf, the larvae received the first dose of MPTP, followed by the second dose at 5 dpf. At 6 dpf, the larvae were collected for further analysis.

**Figure 2 biomedicines-12-02497-f002:**
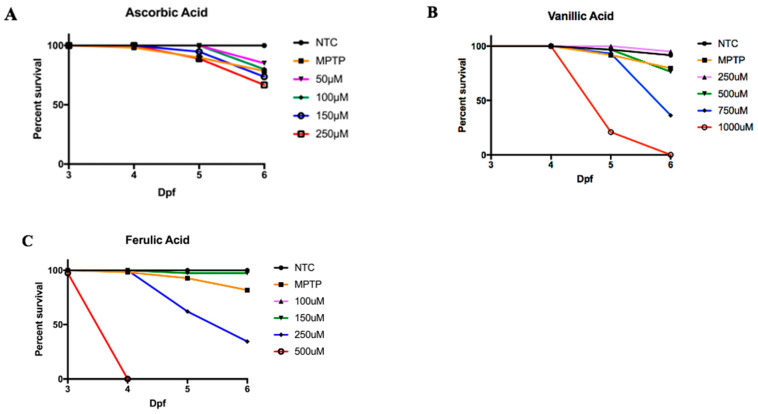
Zebrafish larvae survival following exposure to MPTP with pre-treatment with (**A**) ascorbic acid (**B**) vanillic acid, and (**C**) ferulic acid at various concentrations to determine the dose to be used in further experiments. Neuroprotection treatments were conducted from 1 dpf to 3 dpf. At 4 dpf, embryos were exposed to 0.25 mM of MPTP concentrations and lethality was quantified every 24 hpt until the zebrafish reached 6 dpf.

**Figure 3 biomedicines-12-02497-f003:**
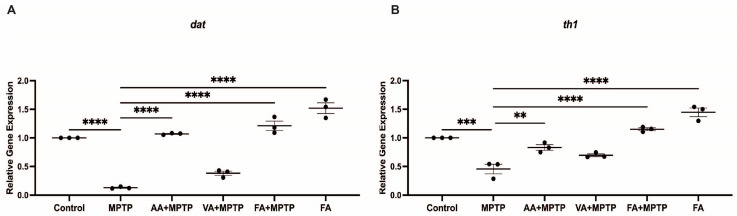
Effects of neuroprotective compounds on *th1* and *dat* gene expression. Relative gene expression of (**A**) *dat* and (**B**) *th1 at* analyzed between control, MPTP, ascorbic acid, vanillic acid, and ferulic acid (n = 3 pools of 20 larvae). One-way ANOVA with Tukey’s multiple comparisons test. ** (*p* < 0.01), *** (*p* < 0.001), **** (*p* < 0.0001).

**Figure 4 biomedicines-12-02497-f004:**
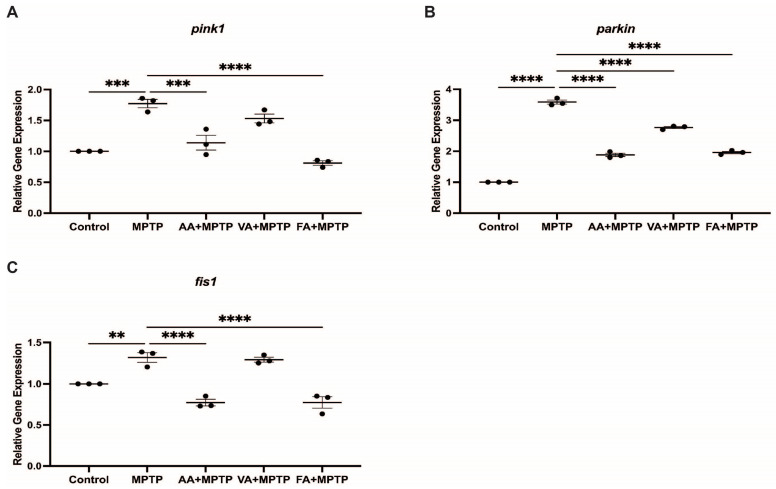
Effects of natural phenolic compounds on mRNA level of genes related to mitochondrial fission. Relative gene expression of (**A**) *pink1*, **(B**) *parkin*, and (**C**) *fis1* analyzed between control, MPTP, ascorbic acid, vanillic acid, and ferulic acid (n = 3 pools of 20 larvae). One-way ANOVA with Tukey’s multiple comparisons test. ** (*p* < 0.01), *** (*p* < 0.001), **** (*p* < 0.0001).

**Figure 5 biomedicines-12-02497-f005:**
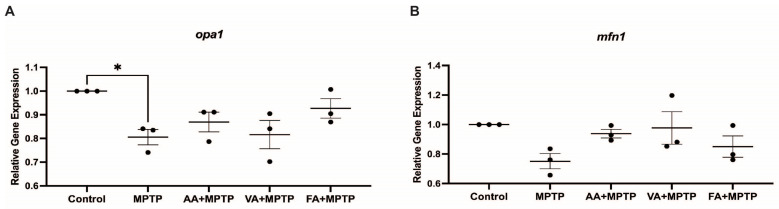
Effects of natural phenolics compounds on mRNA level of genes related to mitochondrial fusion. Relative gene expression of (**A**), opa1 and (**B**) *mfn1* analyzed between control, MPTP, ascorbic acid, vanillic acid, and ferulic acid (n = 3 pools of 20 larvae). One-way ANOVA with Tukey’s multiple comparisons test. * (*p* < 0.05).

**Figure 6 biomedicines-12-02497-f006:**
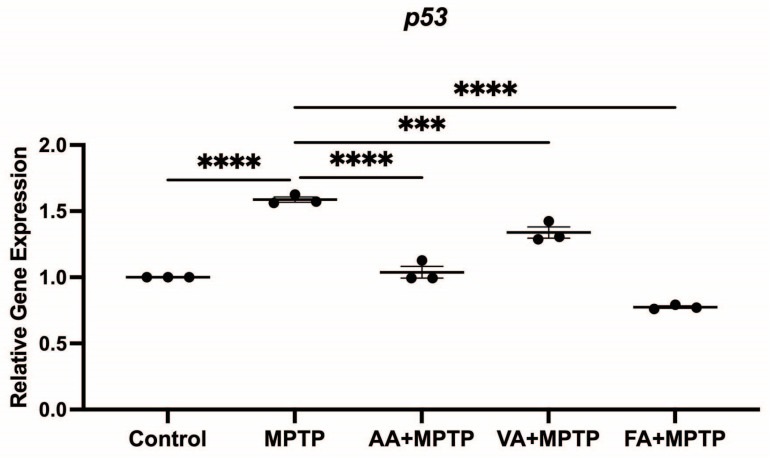
Effects of neuroprotective compounds on mRNA level *p53* gene (related to apoptosis). Relative gene expression of *p53* analyzed between control, MPTP, ascorbic acid, vanillic acid, and ferulic acid (n = 3 pools of 20 larvae). One-way ANOVA with Tukey’s multiple comparisons test. *** (*p* < 0.001), **** (*p* < 0.0001).

**Figure 7 biomedicines-12-02497-f007:**
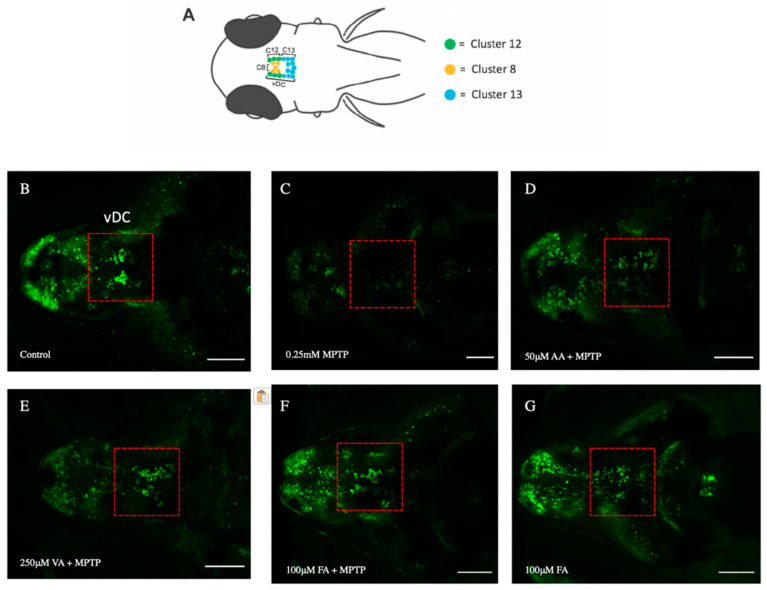
Effects of MPTP, ascorbic acid, vanillic acid, and ferulic acid on three DAnergic clusters located in the vDC of 6 dpf Tg(*dat*:eGFP) zebrafish larvae. (**A**) Schematic representation of eGFP+ neuronal clusters 8 (yellow), 12 (green), and 13 (blue) within the vDC, captured from a dorsal view oriented with the anterior facing the left. (**B**–**G**) eGFP+ cells within the vDC (red boxes) through confocal fluorescence imaging following exposure to 0.25 mM MPTP, 50 μM ascorbic acid, 250 μM vanillic acid, and 100 μM ferulic acid (n = 10–15). All images were acquired using a maximum projection of 2 μm z-stacks, scale bar = 100 μm.

**Figure 8 biomedicines-12-02497-f008:**
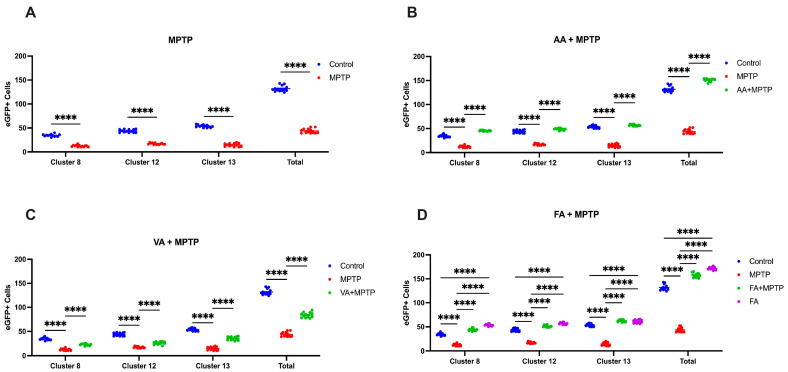
Quantification of DAnergic neurons within the vDC of zebrafish larvae. Developing Tg(*dat:eGFP*) larvae were exposed to (**A**) 0.25 mM MPTP, (**B**) 50 μM ascorbic acid, (**C**) 250 μM vanillic acid, and (**D**) 100 μM ferulic acid (n = 10–15) and live imaged using a confocal microscope to capture vDC DAnergic clusters 8, 12, and 13. (**A**): Two-way ANOVA with Sidak’s multiple comparisons test. (**B**–**D**): Two-way ANOVA with Tukey’s multiple comparison. **** (*p* < 0.0001).

**Figure 9 biomedicines-12-02497-f009:**
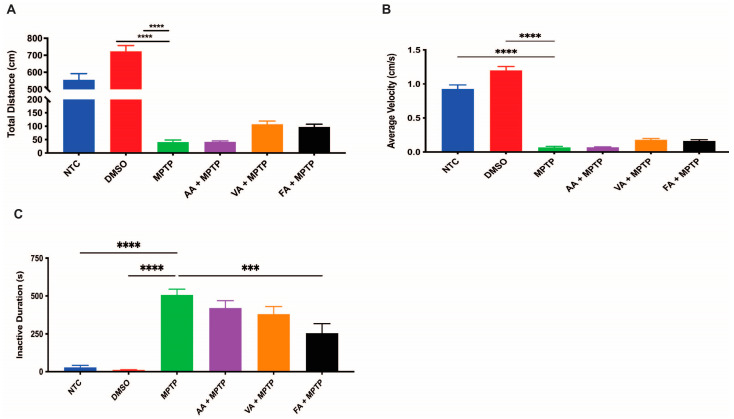
Restorative effects of ascorbic acid, vanillic acid, and ferulic acid on behavior. The swimming activity of 6 dpf larvae (n = 20) in NTC, DMSO, MPTP, ascorbic acid, vanillic acid, and ferulic acid solutions was assessed for (**A**) distance, (**B**) average velocity, and (**C**) inactivity duration. Bars represent the mean ± the SEM. One-way ANOVA with Tukey’s multiple comparisons test. *** (*p* < 0.001), **** (*p* < 0.0001).

**Table 1 biomedicines-12-02497-t001:** List of primers designed for qRT-PCR.

Primer	Forward Sequence (5′–3′)	Reverse Sequence (5′–3′)
*th1*	GACGGAAGATGATCGGAGACA	CCGCCATGTTCCGATTTCT
*dat*	AGACATCTGGGAAGGTGGTG	ACCTGAGCATCATACAGGCG
*pink1*	GGCAATGAAGATGATGTGGAAC	GGTCGGCAGGACATCAGGA
*parkin*	GCGAGTGTGTCTGAGCTGAA	CACACTGGAACACCAGCACT
*fis1*	CCCTGAACCTTCCAGTGTTT	GTCTCTGGAAACGGGTCCTT
*mfn1*	CTGGGTCCCGTCAACGCCAA	ACTGAACCACCGCTGGGGCT
*opa1*	GCTTGAGCGCTTGGAAAAGGAA	TGGCAGGTGATCTTGAGTGTTGT
*p53*	ATATCCTGGCGAACATTTGG	ACGTCCACCACCACCATTTGAAC
*rpl13a*	TCTGGAGGACTGTAAGAGGTATGC	AGACGCACAATCTTGAGAGCAG
*ef1a*	CTGGAGGCCAGCTCAAACAT	ATCAAGAAGAGTAGTACCGCTAGCATTAC

## Data Availability

Data are available from the authors upon request.

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
