# Peer review of "Neuroprotective Effects of Ascorbic Acid, Vanillic Acid, and Ferulic Acid in Dopaminergic Neurons of Zebrafish"

_biomedicines, 2024, doi:10.3390/biomedicines12112497_

Round 1
Reviewer 1 Report
Comments and Suggestions for Authors
Comments to the authors
It is better to use Zebrafish as a keyword instead of Danio rerio
The pathological hallmark of PD alpha-synuclein and Lewy body was not introduced.
“With midbrain DAnergic neurons being so energetically demanding, impacts to mitochondrial dynamics, such as imbalances in fission and fusion, are linked to PD and other neurodegenerative diseases the rely on midbrain DAnergic output” Please correct the last part of this sentence. Several other grammatical errors can also be corrected.
Was there any ethical approval number to conduct this animal study?
What was the rationale for these specific VA, FA, and AA dose selections? How can the doses be represented as mg/kg body weight?
The authors detailed and emphasized the antioxidant properties of these potential drugs without assessing antioxidant activity in the study.

The English could be improved to more clearly express the research.
Author Response
We thank the Reviewer for the helpful comments.
It is better to use Zebrafish as a keyword instead of Danio rerio. Answer: This was corrected.
The pathological hallmark of PD alpha-synuclein and Lewy body was not introduced. Answer: We agree with the reviewer that alpha-synuclein and Lewy bodies are important pathological of PD. However, zebrafish do not have an alpha-synuclein gene (they have 3 synuclein genes, one beta and two gamma) and while some experiments led to protein aggregates, they were not described as Lewy bodies, to he best of our knowledge. We felt that introducing these concepts would distract the reader from the main theme of the manuscript and this is why we did not introduce nor discuss them.
“With midbrain DAnergic neurons being so energetically demanding, impacts to mitochondrial dynamics, such as imbalances in fission and fusion, are linked to PD and other neurodegenerative diseases the rely on midbrain DAnergic output” Please correct the last part of this sentence. Several other grammatical errors can also be corrected. Answer: this sentence was corrected (lines 42-44) and we went through the manuscript again to correct grammatical errors.
Was there any ethical approval number to conduct this animal study? Answer: The ethical approval number (BL-3539) is indicated (Line 425).
What was the rationale for these specific VA, FA, and AA dose selections? How can the doses be represented as mg/kg body weight? Answer: The range of concentrations that were used are based on previous studies. This is now indicated on Lines 95-97. Unfortunately, as zebrafish larvae weigh less than a mg and as the compounds are added directly to tank water, it is not possible to express doses as mg/kg body weight. This is why they were presented as molar concentrations.
The authors detailed and emphasized the antioxidant properties of these potential drugs without assessing antioxidant activity in the study. Answer: Unfortunately, it would be difficult to assess antioxidant properties of the compounds in live zebrafish larvae because the very little amounts of available tissue. Fortunately, the antioxidant properties of these compounds are well established as described in both the Introduction and Discussion.
Reviewer 2 Report
Comments and Suggestions for Authors
This study shows that the pretreatment of ascorbic acid (AA), vanillic acid (VA) or ferulic acid (FA) exerts positive neuroprotective effects in dopaminergic neurons of PD model zebrafish induced by MPTP. The experiments were performed adequately and the story seems according to the experimental results. The three drugs were used for the study. They are drugs that have been studied about their neuroprotective effects well before. They had various potencies in each experiment. Is there any correlation among those drugs in each test? What differences of affecting mechanisms of the three drugs induce those differences? The author should explain those differences among the three drugs in “Discussion”. I have some minor concerns.
1. In Fig.8D, the effects of FA alone should be compared with “Control” .
2. In p13, L9, “FA-treated larvae had the highest percentage of active larvae (60%)”, please explain what percentages are more specifically.
3. In p15 and p16, the numbers of subtitles are 4.2 and 4.3 instead of 3.4.2 and 3.4.3.
4. In p16, L1, at the end of the sentence, the reference number [49] might be put down.
5. In p19, L17, the reference [49] may be wrong. The correct reference should be drawn.
Author Response
We thank the Reviewer for the helpful comments.
This study shows that the pretreatment of ascorbic acid (AA), vanillic acid (VA) or ferulic acid (FA) exerts positive neuroprotective effects in dopaminergic neurons of PD model zebrafish induced by MPTP. The experiments were performed adequately and the story seems according to the experimental results. The three drugs were used for the study. They are drugs that have been studied about their neuroprotective effects well before. They had various potencies in each experiment. Is there any correlation among those drugs in each test? What differences of affecting mechanisms of the three drugs induce those differences? The author should explain those differences among the three drugs in “Discussion”. I have some minor concerns.
Answer: We thank the reviewer for these suggestions. We added a section to the Discussion (Section 4.4) to address this point. Some additional suggestive elements as to mechanistic differences can also be found in the Discussion sections that pertain to each individual compound (Sections 4.1 to 4.3)
- In Fig.8D, the effects of FA alone should be compared with “Control” .
Answer: The figure was revised as suggested.
- In p13, L9, “FA-treated larvae had the highest percentage of active larvae (60%)”, please explain what percentages are more specifically.
Answer: We revised this paragraph to better align sentences with the figure panels and to better emphasize the effects that reached statistical significance.
- In p15 and p16, the numbers of subtitles are 4.2 and 4.3 instead of 3.4.2 and 3.4.3. Answer: this was corrected.
- In p16, L1, at the end of the sentence, the reference number [49] might be put down. Answer: This was corrected. Line: 358
- In p19, L17, the reference [49] may be wrong. The correct reference should be drawn. Answer: This was corrected (Line 375).
Reviewer 3 Report
Comments and Suggestions for Authors
The reviewer would like to declare no conflict of interest with the authors.
1. Extensive use of abbreviation throughout the manuscript. From my understanding, MDPI doesnot impose word limit on the manuscript. The authors introduced many uncommon abbreviations in the manuscript, that do not serve any practical purposes. AA, VA, FA?? Some abbreviations were not properly defined in the first use, eg. ROS.
2. The abstract should be revamped. "VA treatment displayed a slightly positive impact on DAnergic protection..."For a research article, vague description should be avoided. What does it mean by "slightly positive impact"... if there's no statisfical significant difference, then it should be speficied.
3. Statistical analysis. "For comparisons of related observations, a two-way ANOVA was employed".... this sentence has no meaning. What is "related observations"?? how many parameters involved?
4. The description of purpose and the description of results for determination of optimal dose (Fig 2) is confusing. ON what basis that the dose obtained is called optimal dose? base on the non-lethal dose? Non-lethal dose is not equivalent to effective dose.
5. This leads to the question on single dose testing for results on the other tests, which only employ single dose for each test compound, which the justification of dose selection was not well provided. The authors should at least provide the ED50 obtained from the literature as reference.
Author Response
We thank the Reviewer for the helpful comments.
- Extensive use of abbreviation throughout the manuscript. From my understanding, MDPI does not impose word limit on the manuscript. The authors introduced many uncommon abbreviations in the manuscript, that do not serve any practical purposes. AA, VA, FA?? Some abbreviations were not properly defined in the first use, eg. ROS.
Answer: We agree with the reviewer and have used the full name instead of the abbreviation for the three neuroprotective compounds. We have also used the full name for ROS.
- The abstract should be revamped. "VA treatment displayed a slightly positive impact on DAnergic protection..."For a research article, vague description should be avoided. What does it mean by "slightly positive impact"... if there's no statistical significant difference, then it should be specified.
Answer: We agree with the reviewer and have revised he abstract. Thus, the sentence regarding the effects, if any, of vanillic acid was modified, considering the protective effect on DAnergic neuron numbers was modest, even though it reached statistical significance. A sentence on the beneficial effects of ferulic acid on locomotion was removed. Finally, we removed vanillic acid from the concluding sentence that starts with “These findings suggest…..”
- Statistical analysis. "For comparisons of related observations, a two-way ANOVA was employed".... this sentence has no meaning. What is "related observations"?? how many parameters involved?
Answer: The section of Materials and Methods on Statistical analysis has been modified and details were given in the legends of Figures 3-7, 9.
- “The description of purpose…….” and
- “This leads to the question…….. “
Answers to points 4 and 5: The choice of an initial range of concentrations for each of the three compounds was based on previous studies. This is now mentioned on Lines XXXX of Materials and Methods. The literature in aquatic species is more extensive for ascorbic acid than for vanillic acid and ferulic acid. Section 3.1 to better explains how we chose each dose to pretreat embryos. The main criterion was survival rate as high doses of vanillic acid or ferulic acid was toxic. Although we saw the use of the term “optimal dose” in other exposure studies, we removed it from our manuscript. To the best of our knowledge, there are no ED50 for this type of assays in the literature and doses used for rodent cannot be easily adapted to zebrafish because of the small size of the embryos/larvae and the method of administration.
Round 2
Reviewer 1 Report
Comments and Suggestions for Authors
Comments justified. Good luck.
Author Response
I thank the reviewer for the comments and thehelpful review of our manuscrript.
Reviewer 3 Report
Comments and Suggestions for Authors
The MDPI reviewer system does not show the reviewer's previous comments.
The author should at least provide a point-to-point rebutal.
In addressing my previous comments #4 and #5, the authors did not put in any effort to even show the comments. Instead, only excepts were shown.
- “The description of purpose…….” and
- “This leads to the question…….. “
This showed the unprofessionalism of the authors and the arrogance.
The article should be rejected based on the arrogance of the authors, and the high similarity index as detected by the iThenticate software, 42%.
Author Response
I thank the reviewer for the comments. We are truly sorry and apologize if the reviewer saw the way we answered his/her comments as arrogant and we can assure this was not the case. The reviewer can be assured that we had read his/her comments carefully. The only reason we combined and abbreviated the reviewer's comments 4 and 5 in our reply without copying them in their entirety was to keep the reply to him/her short as the reviewer already had comments 4 and 5. We did address them in our revised manuscript.
Finally, we want to point out that the high similarity index is only due to the fact that sections of the manuscript are from the PhD thesis of one of the authors, Michael Kalyn, as his thesis was written using the "on article" format, where articles or manuscript are directly integrated into the thesis, a widely used and accepted practice in many universities, including ours. We than the reviewer for his/her understanding.